# The Past, Present, and Future of Genetically Engineered Mouse Models for Skeletal Biology

**DOI:** 10.3390/biom13091311

**Published:** 2023-08-26

**Authors:** Megan N. Michalski, Bart O. Williams

**Affiliations:** 1Department of Cell Biology, Van Andel Institute, Grand Rapids, MI 49503, USA; megan.michalski@vai.org; 2Core Technologies and Services, Van Andel Institute, Grand Rapids, MI 49503, USA

**Keywords:** transgenic mouse, cre recombinase, bone, osteoblast, osteocyte, chondrocyte, osteoclast, cartilage, CRISPR, *i*-GONAD

## Abstract

The ability to create genetically engineered mouse models (GEMMs) has exponentially increased our understanding of many areas of biology. Musculoskeletal biology is no exception. In this review, we will first discuss the historical development of GEMMs and how these developments have influenced musculoskeletal disease research. This review will also update our 2008 review that appeared in BONEKey, a journal that is no longer readily available online. We will first review the historical development of GEMMs in general, followed by a particular emphasis on the ability to perform tissue-specific (conditional) knockouts focusing on musculoskeletal tissues. We will then discuss how the development of CRISPR/Cas-based technologies during the last decade has revolutionized the generation of GEMMs.

## 1. Introduction

### The Beginning of “Transgenic” Mice

Mouse models of human disease have played critical roles in biomedical research. Forward genetics screens, where phenotypic screening of organisms with spontaneous or randomly induced mutations (for example, following ENU treatment) followed by subsequent identification of the underlying genetic alteration, have contributed to understanding skeletal development and disease [1,2,3,4]. In this review, we focus on the history of using genetically engineered mouse models for reverse genetics in which phenotypic assessment follows the creation of a candidate genetic variant.

“Transgenic mice” are mice in which DNA from another source is introduced into the mouse genome. The first example was in 1974 when mouse embryos infected with the SV40 DNA tumor virus were shown to incorporate the viral DNA into their genome and transmit it to their offspring [5,6]. Subsequent work demonstrated that foreign DNA could be directly injected into the pronuclei of fertilized mouse eggs and integrate into the genome [7]. While these techniques were transformative, there were limitations. The biggest was that the introduced DNA would randomly incorporate into the host genome and multiple copies of the introduced DNA could insert into locations throughout the genome and/or concatemerize and incorporate into these locations. Thus, differences in gene expression from these introduced pieces of DNA could result from different insertion sites.

To overcome these limitations, approaches based on manipulating mouse embryonic stem (ES) cells were developed. ES cells are derived from the inner cell mass of mouse blastocysts (approximately embryonic day 3.5 (E3.5) embryos) [8,9]. These cells retain the pluripotency to differentiate into all tissues of the mature mouse, including the germline. Specifically, ES cells can be injected into a mouse blastocyst to create chimeric mice that could transmit the genetics of the ES cells through the mouse germline [10].

Using ES cells, seminal work during the late 1980s developed methods to alter specific genomic sites within ES cells. In 1987, Oliver Smithies demonstrated that DNA homologous to host DNA would integrate into that site with some level of frequency [11]. Concurrently, Mario Capecchi’s laboratory developed the positive–negative selection method to enrich for these homologous recombination events and allowed virtually any site in the mouse genome in ES cells to be targeted for homologous recombination. One could create ES cells with modified gene locations by designing targeting vectors for homologous recombination, electroporating them into ES cells, and then isolating single-cell clones. By expanding these “correctly” targeted ES cell clones and injecting them into host blastocysts to create chimeric mice, these changes could be transmitted through the mouse germline to establish strains of mice harboring the desired change.

The ability to perform “mouse knockouts” using ES cells was revolutionary, and the impact of this technology was recognized by the Nobel Prize in Physiology and Medicine in 2007. The initial applications of these techniques typically focused on creating mice with deletions (null mutations) in specific genes. These so-called “global” knockouts were instrumental in establishing the physiological functions of many genes. For example, one of the first gene knockouts ever made was in Src, which, surprisingly to many at the time, developed severe osteopetrosis [12]. However, there were limitations to the utility of global knockouts. For example, it became apparent early on that homozygosity for many null alleles results in embryonic lethality [13]. This prevented assessing the effect of specific gene alterations in adult tissues. In addition, the fact that most genes function in several cell types often confounded the interpretation of the cellular mechanisms underlying the phenotypes associated with homozygous for null mutations in mice. The ability to selectively delete genes in specific lineages and tissues was sought for these and other reasons. This was primarily accomplished by applying Cre–lox recombination systems to mouse models.

## 2. Conditional Knockouts for Skeletal Biology

The Cre–*lox* system was identified in bacteria in the early 1980s [14]. The 38 kDa P1 bacteriophage cyclization recombination (Cre) protein catalyzes recombination between two *loxP* sites. *loxP* (locus of X-over P1) sites are 34-base-pair consensus sequences containing a core domain of 8 base pairs flanked on each side by a 13-base-pair palindromic sequence [15]. Cre-mediated recombination deletes the sequences flanked by the *loxP* sites. The utility of this system in eukaryotic cells was first demonstrated in the late 1980s [16,17,18], and further confirmation of its activity in transgenic mice was shown in 1992 [19,20]. This led to numerous mouse strains in which *loxP* sites (so-called “floxed” strains) flanked essential portions on the gene. Cre-mediated recombination creates a null gene if the floxed alleles are appropriately designed. By controlling where and when Cre recombinase is expressed, gene inactivation can be restricted to specific tissues and time points. This has resulted in the creation of hundreds of transgenic mouse strains where Cre is expressed under the control of tissue-specific promoters [21]. Overall, the Cre–lox system has facilitated significant insights into the tissue-specific functions of thousands of genes in mouse models, including within the skeletal system. Transgenic mice that express Cre under the control of specific skeletal promoters are available to direct Cre expression within most significant cell types of the musculoskeletal system, including chondrocytes, osteoblasts, osteocytes, and osteoclasts.

## 3. Cell Types of the Osteochondral Lineage

Osteoblasts and chondrocytes are derived from a common mesenchymal stem cell progenitor (Figure 1). Chondrocytes are differentiated cells that establish a rigid but flexible cartilage extracellular matrix [22]. Bone formation can occur by endochondral or intramembranous ossification (reviewed in [23]). During endochondral ossification, the cartilaginous growth plate establishes the preliminary skeletal structure by which the axial and appendicular skeleton is generated. Mineralized bone is then deposited along this cartilaginous scaffolding in the process of endochondral ossification. The other type of bone formation is intramembranous ossification, where bone is directly formed from mesenchymal progenitors without a cartilage intermediate.

Osteoblasts are responsible for secreting and mineralizing osteoid during bone formation [22]. Once the osteoblast has deposited enough bone to trap itself inside the ossified matrix, it is called an osteocyte. Osteocytes no longer secrete osteoid but regulate osteoblast activity and likely respond to mechanical transduction [24].

Part of maintaining bone homeostasis is controlling mineralized bone and calcium resorption. This process is crucial in establishing a stable serum Ca^2+^ ion concentration and is regulated by osteoclasts, which are multinucleated cells derived from the hematopoietic lineage. Osteoclast-mediated bone resorption occurs at the surface of the bone. It is coordinated with osteoblast activity, in part by the engagement of a receptor activator of nuclear factor kappa-B (RANK) receptor on the osteoclast with RANK ligand (RANKL), secreted from the osteoblast [25]. Osteoblasts also produce osteoprotegerin (OPG), a soluble decoy receptor for RANKL. OPG inhibits the binding of RANKL to its receptor RANK on osteoclasts, thereby suppressing osteoclast formation and activity. The recognition of the intercellular communication systems has facilitated new treatments for bone disease, such as the development of Denosumab, which is a monoclonal antibody that mimics the activity of OPG [26,27]. Interestingly, mouse models played a large role in providing the foundation to develop Denosumab. One of the first groups to identify OPG (at Amgen) did so based on a project where transgenes directing the expression of secreted proteins with unknown functions were created to express these proteins from the liver. In the case of OPG (TNFRSF11B), the resulting transgenic mice had extremely dense bones [28].

We have previously reviewed the various transgenic mouse strains that are available to facilitate Cre-mediated deletions within cell types of the skeleton, and this section of the review will serve to update that work [29]. In Figure 1, we provide a schematic diagram indicating the relative temporal and cell-type-specific expression of the Cre drivers discussed in this review.

## 4. Mesenchymal Stem Cells within the Skeletal System

Bone marrow mesenchymal stem cells (BM-MSCs) are nonhematopoietic, multipotent stem cells that can self-renew or differentiate into cells of several lineages, including adipocytes, osteoblasts, and chondrocytes. BM-MSCs are a desirable target for lineage tracing experiments due to their heritability, and several inducible Cre models have been developed to track various lineages through development, healing, and aging. However, because BM-MSCs give rise to many cell types, BM-MSC Cre lines are less ideal for conditional deletion studies. We summarize several useful BM-MSC-Cre lines that have been instrumental in the understanding of skeletal stem cell lineages.

### 4.1. Gli1

Glioma-associated oncogene homolog 1 (*Gli1*) is a direct transcriptional target and effector of Hedghog (Hh) signaling and is known to be an important regulator of bone and cartilage development. Ahn et al. [30] developed a tamoxifen-inducible *Gli1-CreER^T2^* line for lineage tracing experiments. In the developing mouse, induction of Cre recombinase at embryonic day (E)13.5 showed Cre expressing cells labeled in the perichondrium, which contributed to osteoblasts, adipocytes, and stromal cells in 2-month-old mice [31]. In the same study, postnatal day (P)30 induction of Cre recombination prior to femoral fracture showed *Gli1*-positive cells contribute to the cartilage and bone cells within the healing callus. *Gli1*-positive cells decrease with age and are no longer detectable at 12 months of age, suggesting they are not key players in the healing of fractures in older animals.

### 4.2. Grem1

Gremlin 1 (Grem1) is a Bmp signaling antagonist and is highly expressed in the metaphysis of long bones [32]. Worthley et al. [32] developed the *Grem1-CreERT* line that, when induced with tamoxifen at P1, showed Grem1-positive cells in the growth plate of 1-month animals. Specifically, marrow stromal cells, osteoblasts, chondrocytes, and periosteal tissues were marked with Grem1. Tamoxifen induction at 8 weeks, followed by femoral fracture 1 week later, showed that Grem1-positive cells contribute to chondrocytes and osteoblasts in the callus 1 week post-fracture.

### 4.3. LepR

Leptin receptor (LepR) is expressed in MSCs, which differentiate predominantly down the adipocyte lineage but can also contribute to the formation of some populations of osteoblasts and chondrocytes. The *LepR-Cre* was developed by Defalco et al. [33] for conditional deletion of genes in cells expressing LepR, and it was later used for tracing experiments [34]. *LepR*-positive cells were found to contribute mostly to adipocytes in mice at 2 months of age, and only contribute minimally to the osteoblast population until 6 months of age. After 14 months, *LepR*-positive cells were found to constitute most osteocytes in the long bone. In tibial fracture healing, *LepR*-positive cells are present in high amounts in the cartilage and bone of the callus 2 weeks post-fracture [35].

### 4.4. PDGFR-α

Platelet-derived growth factor receptor alpha (PDGFR-α) is highly expressed in the developing mouse mesenchyme [36]. Two tamoxifen-inducible Cre drivers were developed for the lineage tracing of PDGFRα-positive cells: *Pdgfrα-CreER^T2^* [37] and *Pdgfrα-CreER^TM^* [38]. Wattez et al. demonstrated *Pdgfrα*-positive cells located within fetal bones of the developing mouse vertebrae and skull [39]. O’Rourke et al. expanded these studies to find that approximately 38% of adult mouse bone marrow is *Pdgfrα*-positive [40]. More recently, Xu et al. found that *Pdgfrα*-positive cells can be found in the periosteum and are a source of stem cells for appositional bone growth and are crucial during fracture healing [41].

### 4.5. Additional BM-MSC Cre Drivers

There are several other markers of bone marrow stem cells that have been identified, and relevant Cre drivers have been developed for lineage tracing experiments, many of which are thoroughly reviewed in [42]. Examples include *Acta2(αSMA)-CreER^T2^* [43], *Cxcl12-iCreER^T2^* [44], *Mx1-Cre* [45], and *Nestin-CreER^T2^* [46].

## 5. Osteochondral Progenitor Cre Transgenic Strains

We and others have utilized several transgenic strains that direct Cre expression to osteochondral progenitor cells. Among these are the Prx1-Cre, Dermo1-Cre, and Sox9-Cre strains. Because these promoters are active in cells with the capacity to differentiate into multiple cell lineages, they lack the specificity seen in other Cre lines that target cells later in the differentiation process. Therefore, when using one of these strains to delete a gene of interest, the potential contributions of all cells of the osteochondral lineage should be considered. These strains do, however, have utility in studying broad limb patterning.

### 5.1. Prx1

Cre expression is driven from the 2.4 kb Paired Related Homeobox gene 1 (*Prx1*) enhancer [47]. Analysis of mice in which Prx1-Cre was crossed to a reporter in which phosphatase activity is induced upon Cre expression showed that Cre is strongly expressed in the developing limb buds. On embryonic day 9.5 (E9.5), Cre is active in the forelimb and hind limb mesenchyme but not in the ectoderm. By E16.5, Cre expression occurs throughout the developing mesoderm, including the latissimus dorsi muscle and a subset of cells deriving from periocular mesenchyme. Another note to consider when working with this strain is the observed Cre expression in the germ line at a variable level [47]. An tamoxifen-inducible Cre (*Prx1-Cre-ER^T^*) is available to drive Cre-mediated recombination at relevant time points during limb development [48].

### 5.2. Dermo1 (Twist2)

The basic helix–loop–helix transcription factor, Dermo1/Twist2, is highly expressed in the condensed mesenchyme during skeletal development and other mesoderm tissues during embryogenesis [49]. *Dermo1-Cre* is also expressed in mesenchymal osteochondral progenitor cells [50]. The strain was created by a homologous recombination-mediated knock-in of Cre into the *Dermo1* gene locus, specifically replacing exon 1. This allows for more precise expression of Cre in locations where *Dermo1* is normally expressed, as the endogenous promoter controls expression in the normal chromosomal context. Cre activity in this transgenic was detected as early as E9.5 [51]. Cre was highly expressed in mesodermal tissues early in embryonic development, with very low expression in neural and ectodermal tissues. Later Cre activity was noted in the condensed mesenchyme, from which both osteoblasts and chondrocytes arise. It is important to note that these studies were performed with *Dermo1-Cre* heterozygous mice, as homozygote Cre knock-in mice are not viable due to embryonic lethality. This is because mice homozygous for this allele are functionally null for the *Dermo1* gene, as the endogenous exon 1 is replaced with the Cre recombinase cDNA [50]. It is also important to note that because of the high levels of Cre expression in early embryonic mesodermal tissues, this strain can also be used to conditionally delete genes in other tissues, such as the lung [52,53].

### 5.3. Sox9

Sox9 (SRY-box transcription factor 9) is a member of the Sox family of transcription factors and activates chondrocyte-specific marker genes [54,55,56]. Because of its expression in all chondroprogenitors [57], a *Sox9-Cre* strain was generated to facilitate studies assessing chondrocyte-specific gene deletions [58]. The *Sox9-Cre* mice were created via homologous recombination of a targeting vector into the *Sox9* locus. The vector consisted of a 7.7 kb segment of the *Sox9* gene, with a Cre construct fused into the untranslated region of exon 3. The Cre construct was fused to an internal ribosomal entry site (IRES) and followed by an FRT-flanked PGK-NEO cassette.

To verify the expression of *Sox9-Cre*, homozygous Cre mice were crossed with the R26 reporter line. β-Galactosidase staining indicated that Cre was being expressed as early as E10.5 in the limb bud mesenchyme, and by E13.5, all cells in the cartilaginous primordia and perichondrium were β-Galactosidase positive. By E17, all chondrocytes, as well as perichondrial, periosteal, and osteoblast cells were expressing β-Galactosidase, indicating *Sox9-Cre* is expressed in the precursors of the chondrocyte and osteoblast lineages. Furthermore, β-Galactosidase staining was also noted in tendons and synoviums, indicating the mesenchymal cells expressing *Sox9-Cre* also give rise to tendon and synovial cells. In addition, these studies also showed that cell types from a variety of tissues including cells of the spinal cord, intestinal epithelium, pancreas, and mesenchymal tissue within the testis are all derived from Sox9-expressing cells [58].

## 6. Cre Transgenic Strains for Chondrocytes

Numerous studies have utilized Cre-mediated gene deletion within the chondrocyte lineage to gain important insights into the molecular mechanisms of chondrocyte signaling, with significant emphasis on understanding the regulation of the growth plate. We will review several of these strains.

### 6.1. Collagen II α1

One *Col2α1-Cre* strain was created [59] with a construct consisting of 3 kb of the mouse *Col2α1* promoter region, a modified first exon with a mutated initiation codon, followed by a 3.02 kb segment of the first intron ligated to a splice acceptor and an internal ribosome entry site (IRES), and finally the Cre recombinase coding region with SV40 large T antigen polyadenylation signal. Strains created by very similar methods were also generated [60]. Cre activity was first noted in the notochord and cranial mesenchyme just prior to E9. By E9.5, Cre activity was also noted in the somites and otic vesicles. At E11.5, very strong activity in the perinotochordal condensations and cranial mesenchyme was observed. At E14.5, Cre activity was noted in all cartilaginous elements. Interestingly, some nonspecific activity was seen, particularly in the submandibular glands, along with some mosaicism, as approximately 5% of the chondrocytes did not show evidence of Cre activity [59].

A second transgenic line was created utilizing the rat α1 promoter of type II. Unlike the original mouse-derived *Col2α1-Cre*, the rat *Col2α1-Cre* uses only 1.1 kb of the promoter region for type II collagen [61]. It also includes a splice sequence consisting of a segment from the rabbit β-Globin intron, followed by a nuclear localization signal (NLS), the Cre cDNA, and a polyadenylation signal. Specificity for Cre expression in chondrocytes was further improved by including a chondrocyte-specific enhancer element after the polyadenylation signal. The transgene was incorporated into the mouse genome via microinjection into fertilized eggs.

Confirmation of the Cre expression was performed by crossing the Cre transgenic mice with the R26 reporter line and with antisense Cre riboprobes on hindlimb sections. Whole-mount embryo staining indicated Cre expression throughout the skeleton. Additional interrogation showed Cre expression in the growth plate chondrocytes by E15.5. Furthermore, in situ hybridization analysis with riboprobes identified Cre expression only in the cartilage and not in the connective tissues of the hindlimb. This transgenic Cre line was initially used to create a chondrocyte-specific deletion of *HIF-1α* [61]. Their studies’ results also verified that the rat *Col2α1-Cre* is highly expressed in the chondrocytes of the growth plates with no detectable non-specific expression [61].

Additional *Col2α1-Cre* lines engineered with mouse [62,63] and human [64] *Col2α1* promoters have been generated, and several inducible models for temporal control of Cre-mediated recombination have been made [65,66,67,68], all with varying Cre recombinase expression patterns. These lines are extensively reviewed by Couasnay et al. [69].

### 6.2. Collagen 10α1

Another well-established marker of chondrocyte differentiation and cartilage production is type X collagen, and several *Col10α1-Cre* lines have been generated to specifically study hypertrophic chondrocytes. The first reported *Col10α1-Cre* transgene was created by fusing a 1.0 kb fragment of the Col10α1 promoter to 1.2 kb Cre cDNA, followed by the 2.1 kb *hGH* polyadenylation signal [70]. Cre activity was detected in cartilage primordia on E14.5. Furthermore, Cre mRNA expression was analyzed in varying tissue types. Cre expression was seen in the skeleton and skin, but not in the lungs, liver, or other soft tissues. Detailed analysis of femur sections showed that β-Galactosidase staining was only visible in the hypertrophic chondrocytes, not in resting or proliferating chondrocytes. *Col10α1-Cre* is specifically expressed in the lower hypertrophic chondrocytes of the cartilage lineage, with a small amount of expression in the skin. Another *Col10α1-Cre* showed similar Cre recombinase expression patterns in the hypertrophic zone of the growth plate at E14.5 [71]. Chen et al. created a *Col10a1-Cre* (10 kb) with Cre activity seen one day later in E15.5 hypertrophic chondrocytes of the rib and digits [72], while a transgenic line created using a BAC recombineering technique showed activity in E13.5 femurs and humeri [73]. An inducible *Col10a1-CreERT* has also been generated for chondrocyte lineage tracing experiments [74].

### 6.3. Agc1 (Aka Acan)

Aggrecan (ACAN) is a proteoglycan with high expression in the growth plate and articular cartilage. CreER^T2^ was introduced into the 3′UTR of *Acan* to develop an inducible model (*Agc1-CreER^T2^*) [75]. Cre recombinase expression was detected in the proliferating and hypertrophic chondrocytes, but no activity was found in the perichondrium. *Agc1-Cre* expression is maintained in adulthood, making the model particularly useful for studying growth plate and articular cartilage during aging.

### 6.4. Gdf5

Growth differentiation factor 5 (Gdf5) is expressed in developing joints as well as in adult articular cartilage, making it a useful target for joint-related studies [76]. *Gdf5-Cre* transgenic mice were first developed by Rountree et al. [77]. Embryonic expression pattern studies showed Cre activity in E12.5 proximal limb joints, which extended to distal limb joints at E14.5 and was active in the spine, wrist, and ankle at P0. Non-skeletal sites of Cre recombination was noted in the brain, spinal cord, fat pads, ear cartilage, and hair follicles.

Two tamoxifen-inducible strains were generated for temporal control of Cre recombinase activity. *Gdf5-CreER^T2^* developed by Shwartz et al. [78] showed expression patterns in epiphyseal chondrocytes with induction at E10.5 and in articular chondrocytes with E14.5 induction. Induction at E13.5 of the *Gdf5-CreER^T2^* line developed by Decker et al. [79] demonstrated expression in the joint capsule and synovium. Later induction (E15.5 or E17.5) targeted articular cartilage when measured postnatally.

### 6.5. Prg4

Proteoglycan 4 (Prg4) acts as a lubricant for articular cartilage. The *Prg4-GFP-CreER^T2^* line was created to target the articular cartilage in a tamoxifen-inducible manner [80]. Induction at E17.5 leads to Cre recombination in the superficial articular cartilage, whereas induction at P21 leads to recombination in deeper articular cartilage, synovia, and ligaments. Non-skeletal activity was reported in the heart and liver.

### 6.6. PthrP

Mizuhashi et al. [81] developed the tamoxifen-inducible *PTHrP-CreER^T2^* line using the parathyroid hormone-related protein (PTHrP) promoter. When induced at P6, this line targets chondroprogenitors in the resting zone of the growth plate, which can be traced for a year after induction. These cells differentiate into hypertrophic chondrocytes, columnar chondrocytes, and osteoblasts.

## 7. Osteoblast-Specific Transgenic Cre Strains

As the sole cell type responsible for mineralized bone deposition, studies of osteoblast activity are critical to understanding the network of signaling pathways responsible for initiating mineralization and ossification of bone. While many different Cre lines have been established to target osteoblast-specific recombination, four of the most common are *Runx2-Cre, Osterix1*-driven (*Osx1*) Cre, α1 type I collagen (*Col1α1*) Cre, and *Osteocalcin* (*OC*) Cre, each having a slightly different expression pattern.

During the differentiation process, an osteoblast precursor changes protein expression patterns as it differentiates into a mature osteoblast [82]. Two specific markers along this differentiation pathway that have been well-characterized are *Runx2* and *Osterix1*.

### 7.1. Runx2

Runx2 is a transcription factor that is essential for osteoblast differentiation. The *Runx2-iCre* line was developed by Rauch et al. by inserting a codon-optimized Cre recombinase (*iCre*) at the bone-specific distal promoter (P1) translational start site [83]. Cre recombinase activity was assessed by crossing with a ROSA26 reporter mouse and confirmed recombination in neonatal pups at all sites of endochondral and intramembranous bone formation. Activity was detected in periosteal cells, osteoblasts, and osteocytes, but not in osteoclasts, adipose, or muscle.

### 7.2. Osterix1

*Osx1-Cre* is a Cre transgenic strain that directs Cre expression from the *Osterix1* promoter [84]. The gene construct was inserted via homologous recombination into exon 1 of the *Osterix1* locus. In addition to the Cre expression under *Osterix1* promotional control, the gene construct also included a GFP construct fused to the Cre recombinase for easy reporter detection. Characterization of the expression pattern of *Osx1-Cre* was performed by detecting the reporter GFP. Further verification of the Cre activity was performed by crossing the *Osx1-Cre* mice with the R26 reporter line, allowing for β-Galactosidase expression upon Cre recombination. Both reporter methods confirmed Cre activity in both endochondral and membranous bony elements, consistent with expected Osterix1 expression. Tibial sectioning at E14.5 illustrated Cre expression in the inner bone-forming perichondrium and sporadically in hypertrophic chondrocytes by both LacZ staining and fluorescence microscopy. Further characterization illustrated that *Osx1-Cre* activity was largely restricted to the osteoblast lineage throughout embryonic and early postnatal development. This proved particularly interesting because *Osterix1* is typically expressed in low levels in chondrocytes. Still, the absence of Cre activity in most chondrocytes (except hypertrophic chondrocytes) indicates that low-level Cre expression was insufficient for recombination, or that the gene construct was deficient in the chondrocyte-specific regulatory elements for *Osterix1* [84]. A tamoxifen-inducible *Osx1*-*CreER*^T2^ line was generated by Maes et al. [85], with similar expression patterns to the *Osx1-Cre* strain.

### 7.3. Col1α1

The expression patterns of type I collagen, a major protein in osteoid, were primarily characterized by transgenically incorporating different length fragments of the type I collagen α1 promoter fused to the β-galactosidase reporter [86]. Their studies showed that, with a 0.9 kb promoter fragment, β-Galactosidase expression was low and restricted exclusively to the skin. However, with a 2.3 kb type I collagen α1 promoter fragment, high expression levels were also detected in osteoblasts and odontoblasts. Finally, the 3.2 kb promoter fragment yielded β-Galactosidase expression in the tendon and fascia fibroblasts of the mesenchyme as well as osteoblasts and odontoblasts, with low levels in the skin [87]. Based upon this characterization, transgenic lines that directed Cre expression under different Col1α1 promoter fragments were developed. Romain Dacquin et al. generated a Cre strain under transcriptional control of the mouse 2.3 kb promoter fragment [88]. Furthermore, Liu and colleagues of the Department of Medicine at the University of Connecticut Health Center generated two Cre transgenes under different length segments of the rat *Col1α1* promoter: 2.3 kb and 3.6 kb [89].

The 2.3 kb mouse *Col1α1-Cre* was generated by fusing the promoter fragment to Cre recombinase cDNA, followed by an MT-1 polyadenylation sequence. The gene construct was then incorporated into the mouse genome through pronuclear injection [88]. The 2.3 kb mouse *Col1α1-Cre* was characterized by crossing the Cre mice with the R26 reporter line [51]. LacZ staining at E14.5 showed Cre activity in the skull and all long bone ossification centers, with very light staining in the skin of the face and hands. By E16.5 and at 5 days after birth, LacZ staining was found in all bones of the skeleton, while no other staining was detected in any other tissue. Furthermore, histological studies revealed the staining was unique to osteoblast cells and was not found in chondrocytes or osteoclasts [88].

Each of the rat *Col1α1* promoter Cre lines was generated via similar methods. An initial 2.3 kb promoter vector served as the starting vector. Cre cDNA was isolated and cloned into the vector after the promoter. For the 3.6 kb segment, the next 1.2 kb of the rat promoter was isolated and cloned into the vector immediately following the original 2.3 kb segment. A bGH polyadenylation signal was placed at the end of the construct. Integration was performed through transgenic insertion [89].

Verification of each of the rat *Col1α1* promoter Cre transgenes was performed by crossing the Cre mice with the R26 reporter line [51], as well as mRNA analysis from 6-week-old Cre-positive mice. Northern blots revealed that the 2.3 kb *Col1α1-Cre* was expressed in the long bone and the calvaria, with very low expression in skin, tendon, brain, kidney, liver, and lung, only detectable by overexposure of the film. Similarly, the 3.6 kb *Col1α1-Cre* was highly expressed in the long bone and calvaria, with moderate expression levels in the tendons; very low expression was detected in the brain, kidney, liver, and lung upon overexposure of the film. Additionally, histological studies of both Cre constructs, when crossed with R26 mice, showed Cre expression sufficient for recombination in osteoblasts. Specifically, the 2.3 kb *Col1α1-Cre* was expressed in mature calvaria osteoblasts but not in the less differentiated cells of the suture mesenchyme. In contrast, the 3.6 kb *Col1α1-Cre* was expressed broadly in osteoblast lineage cells in the suture mesenchyme. It was concluded that 2.3 kb *Col1α1-Cre* was more specific for osteoblast recombination, while the 3.6 kb *Col1α1-Cre* targeted a slightly broader mesenchymal Cre expression [89].

Zha et al. [90] generated another rat 3.6 kb *Col1a1-Cre* line using a different polyA sequence. This line demonstrated higher specificity than the one generated by Liu et al. [89]. Cre recombinase activity was detected in calvaria, trabecular bone, and osteocytes, but not in the growth plate cartilage, heart, and brain.

Two inducible *Col1-Cre* lines were generated: mouse 2.3 kb *Col1α1-CreER^T2^* [91] and mouse 3.2 kb *Col1-CreER^T2^* [85]. The 2.3 kb *Col1α1-Cre* showed Cre recombinase activity in E18.5 long bone, calvaria, rib, and vertebra osteoblasts, but not in the heart, lung, liver, or kidney. The 3.2 kb *Col1-CreER^T2^* showed similar Cre activity to the *Osx-CreER^T2^*, with expression in the mandible, calvaria, and ribs, but it had reduced activity in the hindlimbs and no activity in the growth plate.

### 7.4. Osteocalcin

A key marker of mature osteoblast differentiation is the production of osteocalcin, a secreted protein that is thought to play an important role in mineralization and bone formation [82]. Several groups have used variations of the osteocalcin promoter to make a wide variety of Cre transgenics. One of these constructs used a 1.3 kb osteocalcin gene 2 promoter segment to drive Cre expression. Furthermore, they generated an artificial OG2 promoter that consisted of six tandem repeats of osteoblast-specific-*cis*-acting element (OSE) 2 followed by six tandem repeats of OSE1, followed by the Col1α1 TATA box driving a cDNA Cre construct with the MT-1 polyadenylation sequence. OSE1 and OSE2 are naturally found in the OG2 promoter, but it was found that the six tandem repeats seemed to be more efficient in driving Cre expression [88]. Verification with R26 reporter mice showed that the 1.3 kb OG2 promoter-driven Cre was only expressed in bone, while the artificial *OG2-Cre* was expressed in bone and cartilage. Their studies further indicated that the Cre expression was relatively weak and/or nonspecific for osteoblasts, and they published findings that the 2.3 kB *Col1α1-Cre* was the most efficient for targeting osteoblasts [88].

Another *OCN*-*Cre* strain was created using a fragment from the human osteocalcin promoter, followed by an intron for rabbit β-globin flanked by small regions of β-globin exon upstream of Cre. In this case, Cre expression was verified by crossing to the Z/AP double reporter line. Cells in which Cre is not expressed stain positive for β-galactosidase, while cells containing Cre expression stain positive for acid phosphatase. Northern blot analysis was also used to verify Cre expression. RNA samples from different tissues indicated that the *OCN-Cre* was preferentially expressed in calvaria, the femur, and the vertebrae (across the skeleton). Numerical calculations between the number of β-Galactosidase-positive cells in Cre transgenic mice and control mice that were crossed to the Z/AP reporter indicated an excision index of 88.4% of osteoblasts and osteocytes, noting Cre expression in nearly 90% of the targeted cells, with extremely low β-Galactosidase-negative cells in control mice. Phosphatase staining indicated that Cre-expressing cells were not present in the calvaria on E16 but could easily be identified on E17 and 18.5 at the ossification centers. Extensive analysis verified that this human *OCN-Cre* was expressed in high levels with high specificity in mature osteoblast cells (and later in the derived osteocytes) [92]. A tamoxifen-inducible human *OCN-CreER^T2^* line [93] shows Cre expression in trabecular and endosteal osteoblasts and osteocytes when induced at 8 weeks.

## 8. Osteocyte Cres

Once osteoblasts become completely embedded in the bone matrix, they change their expression patterns, stop secreting osteoid, and become more active in regulating formation and resorption by crosstalk with other osteoblasts and osteoclasts. These matrix-embedded cells are called osteocytes. Osteocytes play a critical role in bone anabolic responses to mechanical signals, allowing the skeleton to respond to increased loading by stimulating bone growth in areas under higher levels of strain [94,95,96]. They are also a principle source of RANKL, which plays a central role in coordinating osteocyte/osteoblast-osteoclast interactions [97,98]. Evaluating the effects of mechanical loading in genetically engineered mouse models has played a significant role in advancing our understanding of osteocyte function. One such example is the demonstration that Lrp5-deficient mice do not build bone in response to mechanical load [99].

### 8.1. DMP1

Osteocytes have similar characteristics to odontoblasts, which are cells found in the teeth and are responsible for forming the dentin layer [100]. Dentin matrix protein 1 (DMP1) is a matrix protein that is highly expressed in osteocytes and odontoblasts [101]. To characterize gene function within osteocytes, a Cre line under the transcriptional regulation of the dentin matrix protein 1 (*Dmp1*) promoter was created [100] by fusing the 14 kb mouse *DMP1* promoter sequence to the Cre cDNA. The *Dmp1* promoter consisted of the 10 kb promoter region, followed by exon 1 and intron 1 and the first 17 base pairs of initial noncoding region from exon 2. The 15 kb construct was then microinjected into the fertilized mouse eggs to create a transgenic line. Crossing the *Dmp1-Cre* line with R26 reporter mice indicated strong Cre expression preferentially in the osteocytes of 6-day-old mice as well as some off-target activity in osteoblasts. While this strain has been invaluable in evaluating gene function within osteocytes, caution should be taken when interpreting some phenotypes resulting from crosses to floxed alleles due to its extra-skeletal expression patterns [102]. Another *Dmp1-Cre* line was generated by Bivi et al. [103], which includes 8 kb of the *Dmp1* promoter. Like the 10 kb line generated by Lu et al. [100], the 8 kb *Dmp1-Cre* showed recombination in osteocytes and osteoblasts, as well as non-skeletal tissues including muscle and intestinal MSCs.

Powell et al. [104] created the tamoxifen-inducible 10 kb *Dmp1-CreER^T2^* mouse line. While this line has some leakiness without tamoxifen induction, this strain is a valuable tool to assess gene deletion in osteocytes and has less off-target Cre recombination in non-skeletal tissues.

### 8.2. Sost

Sclerostin (Sost) is expressed in mature osteocytes, but not osteoblasts or bone-lining cells, making it a useful marker for studying osteocytes [105]. Xiong et al. [106] developed the *Sost-Cre* line using a BAC clone to insert Cre recombinase into the mouse *Sost* gene. Cre recombinase activity was detected in osteocytes but showed leakiness in the hematopoietic lineage and brain. When Cre recombination was induced at 2 months of age using the *Sost-CreER^T2^* line, Cre activity was limited to the osteocyte population and did not affect osteoclasts [107].

## 9. Osteoclasts

### 9.1. Myeloid Progenitors: Csf1r, LysM, and CD11b

The colony stimulating factor 1 receptor (Csfr1) [108,109], M lysozyme (LysM) [110], and integrin alpha M (CD11b) [111] promoters can also be utilized to target osteoclast precursors for genetic manipulation. However, they are broadly expressed in cells of the myeloid lineage, making them less specific for osteoclast studies. Because osteoclasts have distinct molecular and functional characteristics compared to other myeloid cells, it is recommended to use Cre drivers with higher specificity to osteoclasts.

### 9.2. TRAP and CtsK

The promoter regions for tartrate-resistant acid phosphatase (TRAP) and cathepsin K (CtsK) have both been used to direct Cre expression in osteoclasts [112]. TRAP is responsible for catalyzing the hydrolysis of a number of esters and anhydrides in the resorption of bone [113], while CtsK is a lysosomal cysteine protease that degrades type I collagen [114].

Chiu et al. developed the first *TRAP-Cre* transgene, which is composed of a 0.62 kb segment of the promoter region of *TRAP* and also includes exons 1B and 1C [112]. *TRAP-Cre* was created from the pTRAP-GFP vector in which the Cre cDNA coding sequence was fused to the promoter region and modified through standard digestion and cloning techniques to generate *TRAP-Cre*. Cre recombinase was active in the long bones, vertebrae, ribs, and calvaria. Nonspecific staining was observed in a few soft tissues, such as the liver and heart. Histological data for *TRAP-Cre* showed β-Galactosidase activity in the osteoclasts of the long bones and proliferating and hypertrophic chondrocytes. Cre expression patterns were highly contingent upon transgenic integration sites, as different donors displayed very different Cre expression. Dossa et al. [115] generated an additional *TRAP-Cre* using a 1.8 kb fragment of the TRAP promoter, which showed similar expression patterns to the 0.62 kb *TRAP-Cre* [112].

*CtsK-Cre* was generated from the pGL3-CK5.0 plasmid, consisting of promoter nucleotides −3359 to +1660 of the *CtsK* gene, which was fused to Cre cDNA and modified as above to generate the *CtsK-Cre* construct [112]. *CtsK-Cre* showed moderate LacZ staining in the long bones, calvaria, and ribs, with low levels observed in a few non-mineralized tissues such as the liver. Furthermore, histological analysis of *CtsK-Cre* in the long bones showed osteoclast staining for LacZ. Very few bone marrow cells stained positive, indicating that *CtsK-Cre* is expressed at a later stage in osteoclast development.

Nakamura and colleagues independently made a *CtsK-Cre* line in which the coding sequence of Cre was knocked into the endogenous *CtsK* locus behind the endogenous ATG site using recombineering [116]. This strain was used for studies in which *estrogen receptor α* was specifically deleted in osteoclasts [117]. A limitation of this model is the potential Cre activity in the germline. Additionally, a tamoxifen-inducible *CtsK-CreER^T2^* line was developed by Sanchez-Fernandez et al. [118] using a 3.48 kb portion of the *CtsK* promoter. Induction from E13.5-E17.5 demonstrated recombinase activity in osteoclasts in most of the skeleton. This line also shows Cre activity in the testes, so germline deletion of target genes should be considered when using this line.

## 10. The Future of Mouse Modeling

### 10.1. CRISPR/Cas9

The discovery of the bacterial CRISPR Clustered Regularly Interspaced Palindromic Repeat)/Cas9 system has been one of this century’s most impactful biomedical research events. This system has been adapted for use as an efficient tool for genome engineering [119,120,121,122,123]. It has been simplified so that only two components are required to facilitate basic genomic engineering: the DNA endonuclease Cas9 and single guide RNA(s) (sgRNA) (providing sequence specificity for Cas9-mediated cleavage) [124]. Bioinformatic tools in publicly available databases (for example, CRISPOR http://crispor.tefor.net/ accessed on 24 August 2023) allow for rapid target guide (sgRNAs) identification. sgRNAs require a 20 bp target DNA sequence preceding the protospacer adjacent motif (PAM) sequence of NGG. The likelihood and location of potential off-target events are also accessible in the databases. Individual sgRNAs can be rapidly created, and these sgRNAs and Cas9 mRNA can be generated and purified for embryo microinjection. Because CRISPR/Cas9 microinjection-based techniques allow the type of precise genomic editing that was previously only possible with the more laborious methods associated with creating mouse lines via ES-cell-based genomic engineering, many laboratories quickly adopted them to generate GEMMs.

### 10.2. Prime Editing and Base Editing

The Cas9 endonuclease introduces double-stranded breaks, ultimately relying on non-homologous end joining or homology-directed repair to fix the break, which may introduce random mutations at the target site. Technological advances have been made to allow for genome editing without double-stranded breaks. One approach was to use modified versions of Cas9, which cleave only a single DNA strand at the target site [125]. Base editing and prime editing technologies have also been developed to directly edit the nucleotide composition at the desired site, potentially allowing for more precise editing and fewer unintended edits.

Base editing is a genome editing technology that utilizes RNA-guided programmable nucleases. Komor et al. generated CRISPR/Cas9-cytidine deaminase fusions that directly convert cytidine to uridine and are programmed with a guide RNA [126]. This led to the development of three classes of base editors: cytidine base editors (C to T) [126], adenine base editors (A to G) [127], and cytidine to guanine base editors (C to G) [128]. Base editing requires a PAM sequence and is therefore limited by the ability to identify a unique target sequence with an appropriate PAM within 15–20 bases of the target site.

Prime editing uses a catalytically impaired SpCas9 fused to a reverse transcriptase and utilizes a modified guide RNA to target a desired genomic region [129]. The prime editing guide RNA (pegRNA) provides a template that includes the new sequence to be introduced into the target region. Prime editing is advantageous compared to base editing because a PAM sequence is not required. Both base editing and prime editing are limited by their efficiency. To justify using these systems in vivo, higher efficiencies are needed. Improvements are rapidly being made to both systems to address these concerns.

### 10.3. i-GONAD

While CRISPR-Cas9 technologies permit the creation of specific variants in mice, they still require a large number of animals and expensive microinjection equipment requiring specialized training. *i*-GONAD (improved-Genome editing via Oviductal Nucleic Acids Delivery) accelerates the generation of CRISPR-edited animals [130,131] and reduces the number of animals needed compared to traditional microinjection-based techniques. This is accomplished by eliminating the need for superovulation of oocyte donor females as well as pseudo-pregnant recipient females and their associated vasectomized males. Instead, editing is performed in vivo using wild-type dams on their E0.5–E0.7 embryos. Briefly, pregnant females are anesthetized, their oviduct is surgically exposed, and Cas9, sgRNA, and repair templates (if used) are injected directly into the ampulla where the zygotes reside (Figure 2). Following electroporation of the oviduct, the surgical incision is closed, and normal gestation is resumed. This strategy creates heterozygous, homozygous, and hemizygous mutant alleles, including single nucleotide variants (SNVs) [130,131,132,133,134,135,136,137,138,139]. We have used *i*-GONAD to create several modified alleles and determined that it is ideal for rapidly and efficiently testing VUSs in genes associated with human birth defects, especially those that may lead to embryonic lethality in a heterozygous state.

## 11. Conclusions

Over the past 30 years, generating and characterizing GEMMs has played a central role in advancing our understanding of skeletal development and disease. The accessibility, relative to other species, of mouse ES cells for genetic engineering was a major reason for mice becoming the most studied model for many human developmental processes and diseases. CRISPR/Cas9 systems and the more recent base and prime editing methods are more readily adaptable to any organism for which a well-annotated genomic sequence is available. This has facilitated the generation of additional genetically engineered rat models for skeletal diseases [140,141]. In addition, while large animal models still require more resources and time to establish and maintain, they are more predictive regarding pre-clinical therapeutic modeling [142,143,144,145,146,147]. CRISPR/Cas9 and other recent approaches will undoubtedly accelerate the development of larger animal models for skeletal diseases. However, given the historical importance and large amount of background knowledge of mouse models for skeletal disease, it will likely remain the key organism for studies of skeletal development and disease for the foreseeable future.

## Figures and Tables

**Figure 1 biomolecules-13-01311-f001:**
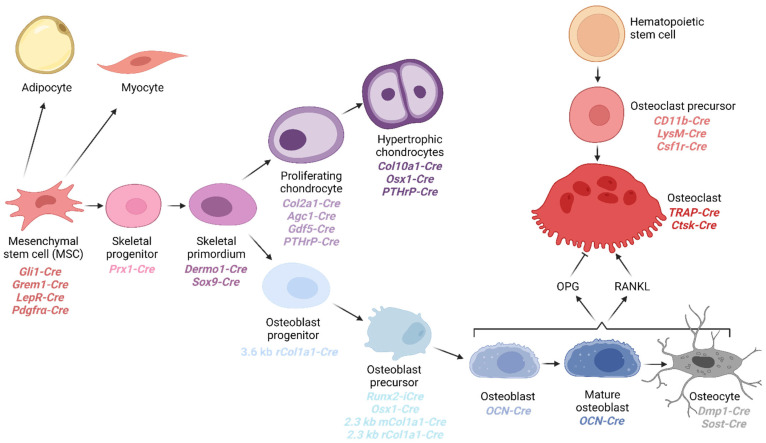
Cells of the mesenchymal and hematopoietic stem cell lineages give rise to the main cells in the skeleton. Mesenchymal stem cells are multipotent stem cells that can differentiate into osteochondral progenitor cells, giving rise to the chondrocyte and osteoblast lineages. Hematopoietic stem cells can differentiate into osteoclast precursors and eventually become multinucleated bone-resorbing osteoclasts. Relevant Cre transgenic mouse strains are indicated for each differentiated cell type.

**Figure 2 biomolecules-13-01311-f002:**
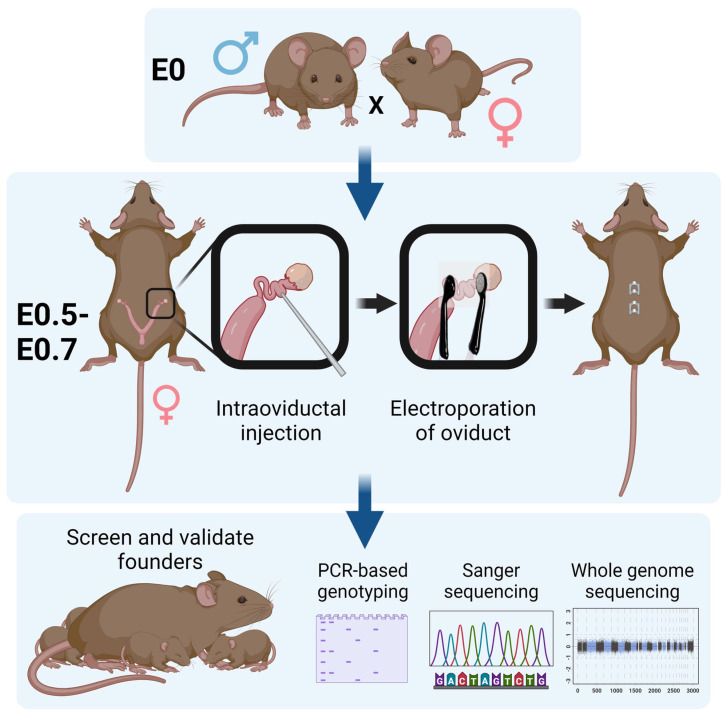
The *i*-GONAD technique is performed on E0.5–E0.7 embryos within the oviduct of a pregnant mouse. CRISPR reagents including Cas9, sgRNA, and repair templates are injected into the oviduct and electroporated. Following surgical incision closure of the female mouse, the edited embryos continue through normal gestation and tissue can be collected from pups to validate genomic editing. Schematic has been modified from Ohtsuka et al. [131].

## Data Availability

No new data were created or analyzed in this study. Data sharing is not applicable to this article.

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
