# Peer review of "The Past, Present, and Future of Genetically Engineered Mouse Models for Skeletal Biology"

_biomolecules, 2023, doi:10.3390/biom13091311_

Round 1

Reviewer 1 Report

In this review, the authors provide a brief historical summary of development of transgenic mice, touch on global knockouts for musculoskeletal tissues, and then turn in detail to tissue-specific (conditional) knockouts and musculoskeletal tissues, with focus on bone and cartilage. They end with a description of the more recent development of CRISPR/Cas-based technologies. This is a “how to” or biotechnology-centered review, i.e., centered on descriptions of approaches to ablating or altering function of specific genes of interest in skeletal cell types of interest rather than a review of the consequences of such modifications. It will be of interest and useful to those entering the bone-cartilage field or embarking on the use of GEMMs for their studies on bone-cartilage. I do think it would benefit from some revisions.

Specific comments

1. The review launches directly into a Section 1 titled The Beginning of “Transgenic” Mice. I think the review would benefit from a preamble or introduction of the broader concept of development of genetically engineered mouse models (GEMMs), mouse models that have had their genomes altered through the use of genetic engineering techniques, i.e., not just transgenic mice, as well as a sentence of what the review will not cover, e.g., large scale mutagenesis approaches to GEMMs.

2. As the authors mention a couple times, the review updates a review from the Williams group in 2008. An update is useful, and the decision not to cover such topics as global knockouts in detail is acceptable given approaches available today. However, it looks like they haven’t updated the section itorducing the osteoblast-osteoclast-chondrocyte lineages and differentiation (Section 3 and Figure 1); as written, the section is somewhat out of date, with numerous general statements supported by older references (<2010) and often too simplistic concepts for what we know today. While citing earlier/historical literature is to be encouraged, some revisions of the text and addition of newer reviews that summarize current understanding of the cells and their functions should be added. E.g., The introduction to development of the skeleton is simplistic and fails to distinguish endochondral and intramembranous ossification, important for some of the markers and promoters discussed. The sentence “Osteoblasts are found on the exterior surface of developing bone.” Is also too simplistic, leaving the impression that osteoblasts reside only the outside of developing bones. L.91-92 – Osteocytes as mechanotransducers and ref 19 should be updated. Osteoclasts-Pg. 3, l.101-109 would also benefit from updating.

3. Figure 1 is very useful, but is introduced rather cursorily in a short sentence. At the least, at the end of Section 3, the authors could draw readers’ attention to the genes/markers delineating the differentiation stages that they then discuss in detail as they describe targeting specific cell types/differentiation stages.

The last sentence at the end of this section (and the Abstract) indicates this is an update of a 2008 review (ref 21) that appeared in “IBMS BONEKey, a journal that is no longer readily available online”. As far as I know, while a few years of BoneKEy (2012-2017) can be accessed in PubMed (and elsewhere) none of the earlier years are accessible online. If the authors know of access to 2008 (or other early years), I suggest providing a link.

4. pg. 10, l. 426-432 – This repeats the same background on osteoclasts as written earlier in section 3 (with addition of ref 77) and can be deleted.

Author Response

Resonses to Specific comments

  1. The review launches directly into a Section 1 titled The Beginning of “Transgenic” Mice. I think the review would benefit from a preamble or introduction of the broader concept of development of genetically engineered mouse models (GEMMs), mouse models that have had their genomes altered through the use of genetic engineering techniques, i.e., not just transgenic mice, as well as a sentence of what the review will not cover, e.g., large scale mutagenesis approaches to GEMMs.

We have now included a few sentences at the beginning outlining how mice have been used for both forward and reverse genetics studies and that this review will focus on reverse genetics.

  1. As the authors mention a couple times, the review updates a review from the Williams group in 2008. An update is useful, and the decision not to cover such topics as global knockouts in detail is acceptable given approaches available today. However, it looks like they haven’t updated the section introducing the osteoblast-osteoclast-chondrocyte lineages and differentiation (Section 3 and Figure 1); as written, the section is somewhat out of date, with numerous general statements supported by older references (<2010) and often too simplistic concepts for what we know today. While citing earlier/historical literature is to be encouraged, some revisions of the text and addition of newer reviews that summarize current understanding of the cells and their functions should be added. E.g., The introduction to development of the skeleton is simplistic and fails to distinguish endochondral and intramembranous ossification, important for some of the markers and promoters discussed. The sentence “Osteoblasts are found on the exterior surface of developing bone.” Is also too simplistic, leaving the impression that osteoblasts reside only the outside of developing bones.

We have updated these sections as suggested by the reviewer.

  1. 91-92 – Osteocytes as mechanotransducers and ref 19 should be updated. Osteoclasts-Pg. 3, l.101-109 would also benefit from updating.

We have included additional text to explain several other known osteocyte functions.

  1. Figure 1 is very useful, but is introduced rather cursorily in a short sentence. At the least, at the end of Section 3, the authors could draw readers’ attention to the genes/markers delineating the differentiation stages that they then discuss in detail as they describe targeting specific cell types/differentiation stages.

We have added text as suggested to provide more introduction to what is shown in Figure 1.

  1. The last sentence at the end of this section (and the Abstract) indicates this is an update of a 2008 review (ref 21) that appeared in “IBMS BONEKey, a journal that is no longer readily available online”. As far as I know, while a few years of BoneKEy (2012-2017) can be accessed in PubMed (and elsewhere) none of the earlier years are accessible online. If the authors know of access to 2008 (or other early years), I suggest providing a link.

We have been unable to find an accessible link to this material (BoneKEy articles prior to 2012).

  1. 10, l. 426-432 – This repeats the same background on osteoclasts as written earlier in section 3 (with addition of ref 77) and can be deleted.

This was consolidated as suggested.

Reviewer 2 Report

This review examines the historical development of GEMMs, with a focus on tissue-specific knockouts in musculoskeletal tissues. It discusses the transformative role of CRISPR/Cas-based technologies. While well-structured, there are suggestions for improvement of comprehensiveness and clarity.

 1.       Inclusion of Additional Transgenic MSC Progenitor Strains: The review could be enriched by incorporating additional information about transgenic mesenchymal stem cell (MSC) progenitor strains present within the skeletal system, such as Gli1 (Glioma-associated oncogene homolog 1), PDGFR-α (Platelet-derived growth factor receptor alpha), and CD271 (Nerve growth factor receptor, NGFR), among others, could be discussed.

2.       Inclusion of a Summarizing Graph: To facilitate reader comprehension, the addition of a graphical representation summarizing the various transgenic strains targeting different cell types is suggested.

Author Response

  1. Inclusion of Additional Transgenic MSC Progenitor Strains: The review could be enriched by incorporating additional information about transgenic mesenchymal stem cell (MSC) progenitor strains present within the skeletal system, such as Gli1 (Glioma-associated oncogene homolog 1), PDGFR-α (Platelet-derived growth factor receptor alpha), and CD271 (Nerve growth factor receptor, NGFR), among others, could be discussed.

We have now discussed several Cre drivers which can be used for lineage tracing assays due to their expression in skeletal MSCs. We have limited our discussion to tissue-specific Cre drivers, of which we are not aware of a published NGFR-Cre line.

  1. Inclusion of a Summarizing Graph: To facilitate reader comprehension, the addition of a graphical representation summarizing the various transgenic strains targeting different cell types is suggested.

As also suggested by Reviewer 1, we have added additional text to provide an introduction and updated the schematic diagram (Figure 1).